# Cytotoxic Activity of Amaryllidaceae Plants against Cancer Cells: Biotechnological, In Vitro, and In Silico Approaches

**DOI:** 10.3390/molecules28062601

**Published:** 2023-03-13

**Authors:** Lina Trujillo, Janeth Bedoya, Natalie Cortés, Edison H. Osorio, Juan-Carlos Gallego, Hawer Leiva, Dagoberto Castro, Edison Osorio

**Affiliations:** 1Grupo de Investigación en Sustancias Bioactivas GISB, Facultad de Ciencias Farmacéuticas y Alimentarias, Universidad de Antioquia, Calle 70 No. 52-21, Medellín 050010, Colombia; 2Grupo Medicina Molecular y de Translación, Facultad de Medicina, Universidad de Antioquia, Carrera 51 D No. 62-29, Medellín 050010, Colombia; 3Facultad de Ciencias Naturales y Matemáticas, Universidad de Ibagué, Carrera 22 Calle 67, Ibagué 730002, Colombia; 4Unidad de Biotecnología Vegetal, Facultad de Ingeniería, Universidad Católica de Oriente, Rionegro 054040, Colombia

**Keywords:** cancer, cytotoxic activity, Amaryllidaceae alkaloids, in silico assays

## Abstract

Cancer is a major cause of death and an impediment to increasing life expectancy worldwide. With the aim of finding new molecules for chemotherapeutic treatment of epidemiological relevance, ten alkaloid fractions from Amaryllidaceae species were tested against six cancer cell lines (AGS, BT-549, HEC-1B, MCF-7, MDA-MB 231, and PC3) with HaCat as a control cell line. Some species determined as critically endangered with minimal availability were propagated using in vitro plant tissue culture techniques. Molecular docking studies were carried out to illustrate binding orientations of the 30 Amaryllidaceae alkaloids identified in the active site of some molecular targets involved with anti-cancer activity for potential anti-cancer drugs. In gastric cancer cell line AGS, the best results (lower cell viability percentages) were obtained for *Crinum jagus* (48.06 ± 3.35%) and *Eucharis bonplandii* (45.79 ± 3.05%) at 30 µg/mL. The research focused on evaluating the identified alkaloids on the Bcl-2 protein family (Mcl-1 and Bcl-xL) and HK2, where the in vitro, in silico and statistical results suggest that powelline and buphanidrine alkaloids could present cytotoxic activity. Finally, combining experimental and theoretical assays allowed us to identify and characterize potentially useful alkaloids for cancer treatment.

## 1. Introduction

Cancer is a major cause of death and a severe impediment to increasing life expectancy worldwide [1]. Breast, prostate, stomach, and uterine cancers are the main types of cancer in terms of the incidence and mortality cases produced [2]. Breast cancer was the most frequently diagnosed malignancy among women worldwide in 2020, and its burden has been growing with a total of 2.3 million new cases (11.7%) [3,4]. By 2040, the burden from this type of cancer is predicted to increase to over 3 million new cases [2]. Among all new cases of all cancers combined (19.3 million), the burden of prostate, stomach, and corpus uteri cancer corresponds to 7.3%, 5.6%, and 2.2%, respectively [3]. In addition, almost 10.0 million cancer-related deaths occurred in 2020 [3,5]. Stomach (0.8 million, 7.7%), female breast (0.7 million, 6.9%), prostate (0.4 million, 3.8%), and corpus uteri (0.1 million, 1.0%) cancer contribute a significant number of cancer-related deaths [3]. Although stomach cancer is a highly heterogeneous disease, it is considered the third cause of death among oncological patients [3,6]. Additionally, an estimate suggests one in six cancer deaths in women was due to breast cancer [2]. Although prostate cancer incidence and mortality rates and trends are decreasing compared to other types of cancer [3,5], this illness remains the leading cause of male cancer death in many countries, especially Africa, the Caribbean, and South America [5,7]. The data provided herein reflect the negative epidemiological impact on public health [8].

Nowadays, chemotherapy, immunotherapy, radiotherapy, and surgery are traditional methods that are still considered effective [9,10]. Some studies have also investigated the effect of combining several of these strategies before the surgical resection of a tumor, in some cases with significant improvement in treatment [11,12,13]. Unfortunately, alone or in combination, these treatment modalities target rapidly dividing cancerous cells and non-malignant cells and thereby induce toxicity [14]. Although chemotherapy is a standard cancer treatment, it has additional disadvantages, such as poor bioavailability of some anti-cancer drugs, and susceptibility to multi-drug resistance, leading to unsatisfactory therapeutic effects [15]. However, chemotherapeutic compounds have remained the frontline choice for advanced-stage malignancies, and several new targets have been identified in different types of cancer [16,17,18]. This recent knowledge could contribute to the search for new, more effective and safer chemotherapy treatments, focusing on different molecular targets involved in anti-cancer activity for FDA-approved anti-cancer drugs and the side effects of such treatments.

In the last 40 years, cancer drug development has led to the discovery of many bioactive compounds from natural resources, of which 46% of anticancer-approved drugs have been provided [19,20]. In addition, alkaloids represent 50% of naturally occurring compounds with pharmaceutical applications, hence their importance in drug research [21,22]. Vincristine and vinblastine (from *Catharanthus roseus*) are used for myeloid leukemia treatment [23]. In the same way, topotecan and camptothecin (from *Camptoteca accuminata*) are used to treat lung, colon, and breast cancer, and homoharringtonine (from *Cephalotaxus fortunei*) to treat chronic myeloid leukemia [19]. Natural product drugs or semisynthetic analog drugs with nitrogen in their structures from microbes (such as bleomycin, peplomycin, romidepsin, and staurosporin), or from marine organism sources (such as aplidine, cytarabine, and trabectedin) are also widely used as potent anti-cancer drugs [19,24]. Several molecules with antitumoral potential are found among the Amaryllidaceae alkaloids [25]. These alkaloids are a distinctive chemotaxonomic feature of the subfamily Amaryllidoideae, belonging to the Amaryllidaceae family, with 636 alkaloids identified to date [26]. These plants are widely distributed in tropical and subtropical zones, especially in Africa and South America [26]. Early historical references mention Hippocrates and Sorano of Ephesus using extracts of *Narcissus poeticus* L., a species of Amaryllidoideae useful to treat uterine tumors [27]. Recently, Amaryllidaceae alkaloids have been isolated and tested as anti-proliferative agents for different types of cancer [28].

With the aim of finding new molecules for chemotherapeutic treatment of epidemiological relevance, we characterized the cytotoxic activities of chemically characterized alkaloid fractions from *Crinum jagus* (J. Thomps.) Dandy, *Caliphruria subedentata* Baker, *Caliphruria tenera* Baker, *Eucharis bonplandii* (Kunth) Traub, *Eucharis caucana* Meerow, *Eucharis formosa* Meerow, *Phaedranassa lehmannii* Regel, *Phaedranassa ventricosa* Baker, *Zephyranthes carinata* Herb., and *Zephyranthes puertoricensis* Traub. in AGS, BT-549, MCF7, PC3, MDA-MB 231, and HEC-1B tumor cell lines and human keratinocytes, HaCat, as a control. Several of these species are native/endemic to Colombia. Due to the low availability of some plant material, some of these plants were collected and subsequently micropropagated through a biotechnological strategy. Finally, the compounds identified in the most active fractions have been docked to target the Bcl-2 family (Mcl-1, Bcl-xL, and Bcl-2) and HK2 enzymes to predict the affinity of small compounds to potential molecular targets.

## 2. Results

### 2.1. Plant Material

The specimens were explored, collected, and herborized. Subsequently, the identification of the material was made using specialized bibliography and comparison with the reference collection deposited in the herbarium of the University of Antioquia (HUA). Table 1 shows the identification by genus and species of the materials and their collection origin.

### 2.2. Establishment of In Vitro Culture

The species *C. subedentata*, *C. tenera*, *E. caucana*, and *E. formosa* were collected in the field and a bulb of each was selected for the micropropagation process. Figure 1 shows the results obtained in the in vitro multiplication of these four materials, and significant differences were found. *E. caucana* showed the lowest multiplication rate corresponding to 1.5 shoots/explant, while the other three species did not show significant differences with a multiplication rate of 2.8 shoots/explant. In the rooting phase, two to three roots were obtained per shoot.

### 2.3. Cytotoxic Activities of the Different Tested Alkaloid Fractions

The cytotoxic effects of the ten alkaloid fractions of Amaryllidaceae species (Table 2) were assessed on breast cancer cell lines BT-549, MCF-7, and MDA-MB 231, gastric cancer cell line AGS, prostate cancer cell line PC3, and uterine cancer cell line HEC-1B. Results were expressed as the percentage of cell viability at 30 µg/mL, considering the American National Cancer Institute (NCI) guidelines for a promising crude extract, which states that a crude extract with IC_50_ less than 30 μg/mL is a candidate for chemical characterization [29,30]. As illustrated in Table 2, except for *E. formosa*, the alkaloidal fractions did not affect the viability in the control cell line HaCat, as the percent cell viability is greater than 90%, showing their low toxicity to the control line. A percentage equal to or below 50% of cell viability was taken as a control point.

*E. caucana* was the only fraction that showed significant results (<50% cell viability) for the MFC-7 cell line. For the rest of the breast cancer cell lines, *Z. puertoricensis* fraction showed the best activity for BT-594 (51.40 ± 1.42%) and *E. formosa* for MDA-MB-231 (54.81 ± 4.61%). It is important to clarify that although *E. formosa* presented anti-proliferative activity against AGS and MDA-MB 231, it was the cell line that presented the lowest viability for the HaCat control line (<90%). However, the best results (lower cell viability percentages) were obtained for *C. jagus* (48.06 ± 3.35%), *C. subedentata* (52.15 ± 2.10%), and *E. bonplandii* (45.79 ± 3.05%) in the AGS cell line. Therefore, using this cell line, IC_50_ values were determined through the MTT assay for some Amaryllidaceae alkaloids such as haemanthamine (13.18 ± 0.47 μg/mL or 43.74 ± 1.56 μM) and lycorine (4.17 ± 0.18 μg/mL or 14.51 ± 0.62 μM). In addition, doxorubicin (5.73 ± 0.80 μg/mL) was used as a positive control.

In the case of the PC3 and HEC-1B cell lines, none of the alkaloidal fractions showed significant cell death results, with survival percentages above 70%. These results suggest that under the conditions in our laboratory, and with an acute treatment for 24 h at a concentration of 30 µg/mL, some alkaloidal fractions are selective and present cytotoxic potential against certain types of cancer cell lines, without compromising the safety in this case, of immortal keratinocytes (HaCat cell line). Interestingly, 80% of the alkaloidal fractions that were active are native/endemic species from Colombia, showing the chemotherapeutic potential of the chemistry of Colombian Amaryllidaceae in molecular targets of pharmacological interest.

### 2.4. Alkaloid Profile of the Different Alkaloid Fractions

All examined alkaloid fractions were chemically characterized by the presence of their different profiling of Amaryllidaceae alkaloids using GC/MS. The analysis of alkaloid identification was performed by comparing the fragmentation pattern reported in the literature and database of Amaryllidaceae alkaloids. Table 3 shows that distribution and abundance depend on both genus and species. A total of 30 alkaloids derived from crinane, galanthamine, and lycorine-type alkaloids were identified. The highest alkaloid content was found in *C. subedentata* (3.925, 1.824, and 1.225 mg/g DW, respectively) and *Z. carinata* (2.980, 2.628, and 2.307 mg/g DW, respectively), with 8-*O*-demethylmaritidine, haemantamine, hamayne, galanthamine, and lycorine alkaloids being the major contributors in *C. subedentata*.

The species *C. jagus* is characterized by the production of an exclusive group of crinane-type alkaloids corresponding to crinine acetate, buphanidrine, crinamine, and powelline, which were identified only in this species. According to the *Eucharis* genus, it was found that *E. caucana* produces a total of 14 alkaloids followed by *E. bonplandii* with 12 and *E. formosa* with ten. This last species is the major producer of crinane, galanthamine, and lycorine-type alkaloids with values of 0.649, 0.933, and 2.400 mg/g DW, respectively. *E. caucana* produces the greatest diversity of alkaloids with the exclusive presence of 6-*O*-methylpretazettine, narwedine, and *N*-formylnorgalanthamine. A total of nine alkaloids were identified in *Phaedranassa* plants, the species with the lowest diversity and production of alkaloids being crinane, galanthamine, and lycorine-type with values of 0.015, 0.019, and 0.119 mg/g DW for *P. lehmannii* and 0.054, 0.015, 0.013 for *P. ventricosa,* respectively. In the alkaloid analysis of *Zephyranthes* species, fourteen alkaloids were identified in *Z. carinata* and eight in *Z. puertoricensis*, with the exclusive presence of galanthine, lycoramine, aulicine, norlycoramine, and assoanine (identified only in *Zephyranthes* species).

### 2.5. Multivariate Analyses of the Cytotoxic Activities of the Alkaloid Fractions

The PCA was performed using information from the Amaryllidaceae species and their alkaloidal composition (Figure 2). The PCA analysis score plot showed that principal components 1 and 2 explain 51.0% of the total variance, and it was possible to satisfactorily differentiate Amaryllidaceae species according to the alkaloid profile. The PCA loading plot (Figure 2) shows that the most important contributors for *C. subedentata*, *C. tenera*, *E. caucana*, *P. ventricosa*, and *Z. carinata* correspond to the crinane-type alkaloids: crinine (X1), haemanthamine (X6), hamayne (X10), and deacetylcantabracine (X11), followed by type-lycorine alkaloids: 11,12-dehydroanhydrolycorine (X19), 5,6-dihydrobicolorine (X13) and galanthine (X30). Similarly, galantamine-type alkaloids such as sanguinine (X14), lycoramine (X27), and galanthamine (X12) are differentiating molecules for these species. These results show that multivariate statistical analysis allows the grouping of Amaryllidaceae species concerning their alkaloid content.

The screening of alkaloid fractions in cancer cell lines showed that the most promising species were *C. Jagus*, *C. subedentata*, *E. bonplandii,* and *E. formosa* in gastric cancer cell line AGS and breast cancer cell line MDA-MB-231, with higher cytotoxic potential in AGS cells. Therefore, the resulting cytotoxic data expressed as percentage values of the cell growth relative to the control were used to develop multivariate statistical analyses; in particular, the supervised partial least square discriminant analysis (PLS-DA) was utilized to establish a correlation between the alkaloid profile and cell viability in gastric cancer cell line AGS (Figure 3). The maximum negative effect on cell growth was caused for crinane-type alkaloids such as crinine (X1), crinane acetate (X3), buphanidrine (X5), crinamine (X7), and powelline (X9) identified in *C. Jagus*, followed by crinine (X1) and 8-*O*-demethylmaritidine (X2) present in *C. subedentata* and *E. bonplandii*. Finally, 8-*O*-demethylmaritidine (X2) was identified in *E. formosa*. Thus, crinane-type alkaloids could be related to possible cytotoxic effects on gastric cancer cells.

### 2.6. Molecular Docking Analysis

Molecular docking analysis is frequently a useful tool for predicting the affinity of small compounds toward potential molecular targets through the study of binding orientation. Table 4 shows the results of the in silico inhibition of the Bcl-2 family (Mcl-1, Bcl-xL, and Bcl-2) and HK2 enzymes by all the alkaloids identified in the most active fractions. Therefore, the ability of the alkaloids to inhibit cell survival through interaction with these proteins is theoretically explained. The affinity scores obtained for selected compounds showed that the proteins have an affinity for compounds with a binding energy between −3.31 to −8.80 kcal/mol. According to the results of the estimated free energy of binding of both controls, the alkaloids with the lowest interaction energy concerning Mcl-1 protein are 11,12-dehydroanhydrolycorine, deacetylcantabricine and norlycoramine. Structural representations of the best conformation of the complexed Mcl-1 with the three tested alkaloids are shown in Figure 4. Similarly, the alkaloids with the lowest interaction energies with Bcl-xL and HK2 proteins are shown in Figure 5 and Figure 6, respectively.

## 3. Discussion

### 3.1. In Vitro Propagation

Amaryllidaceae is one of Colombia’s rarest and most striking plant families and is endangered due to habitat destruction [31]. In vitro plant tissue culture techniques are an alternative for conserving and propagating endangered species [32]. Although in vitro tissue cultures are scarce for Caliphruria and Eucharis species, the reports on the micropropagation of *E. grandiflora* by direct organogenesis showed that from a mother bulb, it is possible to obtain 3.8 shoots/explant and 133 plants in 5 months [33]. In this study, an adequate multiplication rate was achieved for micropropagated species *C. subedentata*, *C. tenera*, *E. caucana*, and *E. formosa*, which allows obtaining more than 1000 plants in 6 months, allowing the initiation of habitat enrichment programs with these species. Additionally, it is necessary to develop protocols to stimulate the production of alkaloids with biological potential, which are necessary for many species of Amaryllidaceae.

### 3.2. Alkaloid Fractions Differed in Alkaloid Profiles and Cytotoxic Activities

Naturally-derived compounds may be an important source of novel effective anti-cancer drugs [19,20,23]. These new chemotherapeutic agents ought to be seen as necessary due to the growing incidence and mortality of the different types of cancer and the development of resistance to conventional anti-cancer drugs [1,2,3]. Recently, Amaryllidaceae alkaloids have been isolated and tested as anti-proliferative agents [28]. However, searching for the types of Amaryllidaceae alkaloids with the greatest anti-cancer potential is convenient. In this work, the crinane, galanthamine, lycorine, and miscellaneous type alkaloids were identified in ten species of Amaryllidaceae. The alkaloid profile of *C. jagus* shows that this species produces an exclusive group of crinane-type alkaloids in combination with lycorine-type alkaloids, which agrees with previous phytochemical reports [34,35,36,37]. *E. bonplandii*, *E. caucana*, *P. lehmannii,* and *Z. carinata* were shown to produce several crinane, galanthamine, and lycorine-type alkaloids [35,38,39], the last with the highest chemodiversity in alkaloids of Amaryllidaceae, which is also in agreement with previous studies [40,41,42]. Furthermore, and to the best of our knowledge, this is the first report of phytochemical data for *C. tenera*, *E. formosa*, *P. ventricosa,* and *Z. puertoricensis*. Figure 2 shows the separation of the ten species of Amaryllidaceae into four groups. In the PCA loading plot, the alkaloids crinane acetate (X3), buphanidrine (X5), crinamine (X7), powelline (X9), anhydrolycorine (X15), deacetylcantabricine (X11), assoanine (X20), norlycoramine (X29) and aulicine (X28) had the greatest contributions to differentiation and could be considered as characteristic compounds of the species *C. jagus* and *Z. puertoricensis*. In general, all these species were characterized by presenting a notable diversity of alkaloids, the vast majority with all types of alkaloids identified, including miscellaneous-type alkaloids. Therefore, the wide diversity of Amaryllidaceae alkaloids (30 alkaloids identified, Table 3) can be considered in the search for potential anti-cancer drugs.

According to the results, gastric cancer AGS cells were the most susceptible to the cytotoxic action of alkaloid fractions, followed by MDA-MB 231 breast cancer cells (Table 2). The purpose of new chemotherapeutic treatments is selectivity against the affected cells, without affecting healthy cells, which is one of the main problems of oncological treatments [43]. Thus, the AGS cells were subjected to further analysis, due to its higher selectivity index, compared to human keratinocytes HaCat cells (control). The GC/MS analysis of the alkaloid fractions with higher bioactivity showed that some of the identified alkaloids were present in most of these alkaloid fractions at different concentrations. These alkaloids were crinine, hamayne, 5,6-dihydrobicolorine, 11,12-dehydroanhydrolycorine, anhydrolycorine, and lycorine. The correlation between the cytotoxic effect and the relative presence of alkaloids in the active fractions could suggest that the cytotoxic activity obtained is due (at least in part) to the presence of crinane and lycorine-type alkaloids, compounds previously reported as potential anti-cancer agents [44,45]. In this study, the potential of crinane and lycorine-type alkaloids against AGS cells was confirmed with the evaluation of haemantamine and lycorine alkaloids against this cell line, with relevant IC_50_ values of 13.18 ± 0.47 μg/mL and lycorine 4.17 ± 0.18 μg/mL, respectively. The suggestion is also supported by the results of PLS-DA analysis (Figure 3), especially for crinane-type alkaloids, the main contributors to the toxicity observed in gastric cancer cells.

In this study, an attempt was made to determine the possible interaction of the Amaryllidaceae alkaloids at the molecular level within the active sites of some molecular targets in gastric cancer, as well as breast, prostate, stomach, and uterine cancers. This in-silico approach should explain, at least in part, the in-vitro assays, which were based on the quantification of cell viability. For this reason, some of the molecular targets highly expressed in a variety of cancer cells and related to this process are those belonging to the Bcl-2 family (Mcl-1 and Bcl-xL) [46], anti-apoptotic proteins, and hexokinase 2 (HK2), which plays a central role in the cellular uptake and utilization of glucose [47]. The anti-apoptotic Bcl-2 proteins protect against cell death by apoptosis and also allow senescent cells to survive [48], and it has been suggested that they might appear in the early phase of carcinogenesis in the stomach [49]. While HK-2 is an important molecular target in gastric cancer because it is involved in regulating accelerated glucose uptake during aerobic glycolysis [50], and these cells rely preferentially on aerobic glycolysis for energy generation rather than on mitochondrial respiration [51].

Some alkaloids presented low interaction energy with the Mcl-1 protein (Figure 4; Table 4). Although 11,12-dehydroanhydrolycorine does not exhibit hydrogen bridge-type interactions, the system is stabilized by the presence of π-π stacking and π-alkyl interactions with residues Phe273 and Lys276. Norlycoramine’s inhibition is through hydrogen bridge interactions with residues Arg233, Gln229, Phe273, and His277 and two π-alkyl interactions with Leu232, and Lys276. Deacetylcantabricine interacts with residues Arg233, Gln229, Phe273, and Lys276 through hydrogen bonds, and two interactions with delocalized π-type systems: π-alkyl with Leu232 and π-cation with His277. While the compounds 11,12-dehydroanhydrolycorine, deacetylcantabricine, norlycoramine, and powelline were the alkaloids that presented the lowest interaction energies with Bcl-xL (Figure 5). The stability of 11,12-dehydroanhydrolycorine is explained by the presence of a hydrogen bridge with the residue Tyr101, and interactions π-π stacking and π-alkyl type with Ala104, Ala149, Leu130, Leu108, and Val126 residues. The inhibition activity of the norlycoramine alkaloid is due to the presence of a hydrogen bridge with Ala104, π-π stacking, and π-alkyl interactions with Phe97, Leu108, Val126, Leu130, Val127, and Phe146, and π-cation interaction with the Glu129 residue. Deacetylcantabricine activity is explained by the presence of a hydrogen bridge with Ala142, π-π stacking, and π-alkyl interactions with Phe105, Phe97, Ala104, Leu108, and Val126. Finally, powelline is bound to the Bcl-xL active site through two hydrogen bridge interactions with Ala142 and Ser145 residues, π-π stacking, and π-alkyl interactions with Leu130, Ala104, Phe97 and Tyr101. In the systems related to the controls, obatoclax, an anti-apoptotic BCL-2 family inhibitor [52,53], shows π-π stacking and π-alkyl interactions with previously described residues such as Leu130, Phe97, and Ala104. In addition, this molecule shows hydrogen bridge interactions with Ala142 and Ser 145. Doxorubicin exhibits π-π stacking and π-alkyl type interactions with previously described residues such as Val126, Ala104, Phe97, and Ala142. However, this molecule binds differently to the active site through hydrogen bridge-type interactions with Leu139, Arg132, Asp133, and Arg139 residues.

Concerning the interaction with HK2 protein, 6-*O*-methylpretazettine is stabilized by six hydrogen bridge-type interactions with Gln608, Val655, Phe602, Phe604, Pro513 and Ser603 (Figure 6). Additionally, this compound shows π-alkyl interactions with Pro605 and Cys606 residues, as well as a π-cation interaction with the amino acid Glu708. Stabilization of pseudolycorine occurs due to the presence of six hydrogen bonds with Ser603, Phe604, Asp657, Asn656, Asn683, and Gln739. In addition, there are two π-cation interactions with the amino acids Glu708 and Glu742. Molecular docking analyses for this system did not detect π-π stacking or π-alkyl interactions. Concerning buphanidrine, HK2 protein active site stabilization occurs as follows: five hydrogen bonds with Phe604, Phe602, Ser603, Asn706, and Pro513, three π-alkyl type interactions with Leu512, Cys606, and Pro605; and finally a π-cation interaction with Glu708. Powelline is also stabilized by the presence of two hydrogen bonds with Phe604 and Phe602 amino acids and two π-alkyl interactions with Pro605 and Cys606 residues. Obatoclax is stabilized in the active site of the protein by the amino acid Cys606, through hydrogen bonding with different parts of the structure. Additionally, π-π stacking and π-alkyl interactions are formed by Leu 512 and Lys 510, while π-cation interactions are established through amino acid Glu 708. Finally, doxorubicin stabilization in the active site is achieved through hydrogen bonding interactions with Gln 739, Cys 606, Ser 603, and Val 655, and two π-cation interactions with Glu 708 and Glu 742 residues.

Considering the results of molecular docking and the PLS-DA analysis, the alkaloids buphanidrine and powelline present in *C. jagus* belong to the group of alkaloids that could be responsible for the cytotoxic potential in gastric cancer AGS cells (Figure 3) due to the formation of hydrogen bridges and π interactions with the amino acid residues of HK-2 and Bcl-xL enzymes (Figure 5 and Figure 6). Of course, other minor alkaloids (identified or not yet identified) could also act synergistically with cytotoxic activity. However, although these data are theoretical, it has been reported that powelline has antitumor and anti-cancer activity in in vitro models [54]. Several mechanisms of interest to explain the cytotoxic effect of Amaryllidaceae alkaloids have been explored. However, their ability to selectively induce apoptosis (or programmed cell death) in cancer cells has been found to be a key finding for many cytotoxic Amaryllidaceae alkaloids [27,44,45]. Surprisingly, this mechanism has not been addressed in many Amaryllidaceae alkaloids [55]. However, these reports show the potential of Amaryllidaceae alkaloids for the treatment of several types of cancer. More experiments are required to clarify the activity related to other alkaloids such as buphanidrine. To our knowledge, there are still no reports on its cytotoxic activity in gastric cancer. Therefore, this first theoretical insight provides a guide for crinane-type alkaloids which have emerged as interesting targets for cytotoxicity-based [25].

## 4. Materials and Methods

### 4.1. Plant Material

The plant material of the wild species of Amaryllidaceae was collected in Colombia, in the departments of Antioquia, Cauca, Chocó, Nariño, Quindío, and Tolima. The cultivated species were donated by the botanical garden “José Negret” located in Popayán, Colombia, and micropropagated by plant biotechnology at the Plant Biotechnology Unit, Faculty of Engineering, Universidad Católica de Oriente. Rionegro, Colombia (Table 1). One specimen of each species was deposited in the Herbarium of the University of Antioquia (HUA), Medellín, Colombia. The species studied were collected with authorization from the Ministry of Environment with a genetic resource access contract #328.

### 4.2. Micropropagation of Amaryllidaceae Species

The species corresponding to *C. subedentata, C. tenera, E. caucana*, and *E. formosa* are determined as critically endangered and their availability is minimal; for this reason, they were propagated by using in vitro plant tissue culture techniques, while the other species were propagated under nursery conditions using bulb culture. Standardized laboratory techniques were used for in vitro propagation, which consisted of harvesting the bulbs, which were washed with detergent and running water. Then, under aseptic conditions, they were disinfected with 70% ethanol (Sigma–Aldrich, St. Louis, MO, USA) for 2 min and then with 1% sodium hypochlorite (commercial grade, Medellin, Colombia) for 20 min. The modified “twin-scales” technique [56] was used to obtain the explants. Explants were inoculated in MS “Murashige and Skoog” culture medium (PhytoTech Labs^®^, Lenexa, KS, USA) [57], supplemented with sucrose (30 g/L, Sigma–Aldrich, St. Louis, MO, USA), growth regulators, and agar (2.4 g/L, Sigma–Aldrich, St. Louis, MO, USA) under total darkness conditions at a temperature of 24 ± 3 °C. For multiplication, MS medium supplemented with benzyladenine (BAP, 1 mg/L, PhytoTech Labs^®^, Lenexa, KS, USA) and naphthaleneacetic acid (NAA, 0.2 mg/L, PhytoTech Labs^®^, Lenexa, KS, USA) was used, and rooting was carried out in MS medium with activated charcoal (0.1%, PhytoTech Labs^®^, Lenexa, KS, USA). Plantlets were acclimatized under nursery conditions.

### 4.3. Extraction of Alkaloids

Bulbs of wild and micropropagated plants were collected, washed, cut into small portions, and dried at 40 °C for 48 h. The dehydrated bulbs were extracted with methanol (Sigma–Aldrich, St. Louis, MO, USA): water (1:1) by extraction assisted by bath ultrasound at a temperature of 40 °C for 30 min [35,40]. The crude extract was obtained by removing the solvent by evaporation at low pressure and then dissolved in 200 mL of methanol: water (1:1) with H_2_SO_4_ (2%, pH 2, Sigma–Aldrich, St. Louis, MO, USA). Neutral compounds were removed with hexane (3 × 50 mL, Sigma–Aldrich, St. Louis, MO, USA), and the accusation phase was based on NH_4_^+^OH^−^ (25%, pH 10, Sigma–Aldrich, St. Louis, MO, USA). The alkaloids were extracted from the aqueous phase with chloroform (3 × 50 mL, Sigma–Aldrich, St. Louis, MO, USA), and the fraction of alkaloids obtained was stored at 4 °C until the time of analysis.

### 4.4. Alkaloid Analyses by GC/MS

Alkaloid analyses were carried out by GC/MS, according to the methods described by Cortes et al. [40]. For the analysis, we used an Agilent 7890 A GC equipment (Agilent, Wilmington, NC, USA) operating in EI mode at 70 eV with autosampler PAL3 Control SW, detector MS 5975C (SCAN analysis by a quadrupole). A capillary column HP-1 (30 m × 0.250 mm × 0.25 μm, Agilent J&W, Palo Alto, CA, USA) PDMS (phenyl polydimethylsiloxane) was used. The oven temperature was initially set at 80 °C (2 min), increased from 80 °C to 210 °C to 15 °C/min, 210–260 °C to 8 °C/min, 260–300 °C to 15 °C/min, and 4 min at 300 °C. The injection of 1 μL of the sample was performed at 280 °C in splitless mode. The flow of carrier gas (helium) was 1 mL/min. The process of identification and quantification of alkaloids was done by spectral deconvolution using the software Agilent MassHunter Qualitative and Quantitative Analysis version B. 07. 00.

#### Data Processing and Analysis

The alkaloids were identified by comparison of fragments of the mass spectra of the Amaryllidaceae alkaloid spectrotheque of the AgroBio Institute (Sofia, Bulgaria) and information from scientific databases. The compounds were also identified by comparing the mass spectral fragmentation of the compounds with standard reference spectra from the NIST 17 database (NIST Mass Spectral Database, Gaithersburg, MD, USA). The compounds’ Kovats Retention Index (RI) was recorded with a standard calibration mixture of n-hydrocarbons (C7-C40, Sigma–Aldrich, St. Louis, MO, USA). The percentage of TIC (total ion current) was estimated for each alkaloid. The abundance of each compound was calculated using codeine (50 μg/mL) as an internal standard. The area of GC/MS peaks depends on the concentration of the related compounds and the intensity of their mass spectral fragmentation.

### 4.5. Cell Viability Screening of Alkaloid Fractions

#### 4.5.1. Cell Culture

Cell viability screening of alkaloid fractions of the ten species of Amaryllidaceae was performed in gastric cancer cell line AGS (CRL-1739^™^), prostate cancer cell line PC3 (CRL-1435^™^), breast cancer cell lines BT-549 (HTB-122^™^), MCF7 (HTB-22^™^) and MDA-MB 231 (HTB-26^™^), uterine cancer cell line HEC-1B (HTB-113^™^) and human keratinocytes HaCat (PCS-200-011^™^) as a non-carcinogenic control cell line (all cells were obtained from ATCC^®^) [58,59]. The cells were cultured in Dulbecco Modified Eagle Medium-high glucose (DMEM, Sigma–Aldrich, St. Louis, MO, USA) and Ham’s F-12 nutrient medium (F12, Sigma–Aldrich, St. Louis, MO, USA) supplemented with 10% heat-inactivated FBS (Sigma–Aldrich, St. Louis, MO, USA), 100 U/mL penicillin (Sigma–Aldrich, St. Louis, MO, USA) and 100 μg/mL streptomycin (Sigma–Aldrich, St. Louis, MO, USA), 2 mM L-glutamine (Sigma–Aldrich, St. Louis, MO, USA) in a humidified atmosphere of 5% CO_2_ and 95% air at 37 °C. Cells were monitored on a Nikon Eclipse TS100 (Melville, NY, USA) inverted phase contrast microscope. Cell viability experiments were performed when cells reached 75–80% confluence using 0.25% trypsin 1 mM EDTA (Sigma–Aldrich, St. Louis, MO, USA).

#### 4.5.2. ATP Quantification by Bioluminescence

Cell viability was determined by bioluminescence assay, based on the quantification of ATP present, as an indicator of metabolically viable cells [58]. The cells were seeded on 96-well plates with a cell density of 2 × 10^4^ cells/well for 24 h. After incubation, the medium was changed, and the cells were treated with the alkaloid fraction at a concentration of 30 µg/mL, according to the cytotoxic activity criteria for a crude extract as established by the American National Cancer Institute (NCI) in IC_50_ < 30 µg/mL. This criterion for a crude extract has recently been used in research as a screening parameter [29,30], making it possible to reduce the number of experiments in preliminary tests. After 24 h of incubation, 100 µL of CellTiter-Glo^®^ cell viability reagent (Promega, Madison, WI, USA) was added to each well. The plate was left agitating for 5 min to induce cell lysis, and the plate was incubated at room temperature for 15 min to stabilize the luminescent signal. Relative luminescence units were measured using the GloMax 96 microplate luminometer (Promega, Madison, WI, USA). Cell viability was expressed as a percentage of control values set at 100%.

#### 4.5.3. Mitochondrial MTT Reduction

Cells viability was also determined using 3-(4,5-dimethylthiazol-2-yl)-2,5-diphenyltetrazolium bromide (MTT, Sigma–Aldrich, St. Louis, MO, USA) assay. Lycorine (Sigma–Aldrich, St. Louis, MO, USA) and haemanthamine compounds were evaluated at different concentrations (100, 50, 30, 20, 10, 1 μg/mL) in the cell line with the lowest percentage of cell viability (AGS) and incubated at 37 °C with 5% CO_2_ for 48 h. Doxorubicin (Sigma–Aldrich, St. Louis, MO, USA) was used as a positive control (10, 5, 2.5, 1.25, 0.62, 0.31 µg/mL). Subsequently, 50 μL of MTT (1.0 mg/mL buffered saline phosphate) was added. After 4 h of incubation at 37 °C, the MTT culture medium was removed and replaced with 150 μL DMSO (Sigma–Aldrich, St. Louis, MO, USA) to dissolve the formazan crystals. Plates were incubated in dark with agitation for 2 h. Optical density was determined at 540 nm using a microplate reader. All experiments were conducted with three independent trials, each with six replicates.

### 4.6. In-Silico Analysis

The molecular docking analysis of the Amaryllidaceae alkaloids has been carried out using the Autodock 4.2 program [60]. The tridimensional chemical structures of Amaryllidaceae alkaloids and the positive control, obatoclax, and doxorubicin, were downloaded from the PubChem database and were edited using the Maestro program (Schrödinger Release 2022-3: Maestro, Schrödinger, LLC., New York, NY, USA, 2021) available at https://www.schrodinger.com/citations#Maestro (accessed on 1 March 2023), belonging to the Schrodinger suite. In this process, the hydrogen atoms were added, and the protonation states were checked for a pH between 7.0 ± 2.0. Three-dimensional (3D) experimentally known structures of the anti-apoptotic Bcl-2, Bcl-xL, and Mcl-1 proteins were obtained from the Protein Data Bank (PDB), pdb codes: 2W3L [61], 2YXJ [62], and 3KZ0 [63], respectively. Likewise, the three-dimensional structural coordinates of the HK2 protein were retrieved from PDB (PDB ID: 5HEX) [64]. The readers can find theoretical analysis of these structures in the literature [65,66,67,68,69]. The protein structures have been prepared using the Maestro program. In this process, the water molecules, ions, and ligands included in the crystallography pdb file were deleted. Additionally, the bond orders were assigned, hydrogen atoms added, missing side chains included, and amino acid protonation states checked. In the molecular docking experiment, the first step corresponded to computing a set of pre-calculated grids of affinity potentials via AutoGrid, to find suitable binding positions for a ligand on a given macromolecule. In this step, a grid box with dimensions of 60 × 60 × 60 Å and centered in the xyz coordinates for each of the proteins: BCL-2 (40.98, 27.23, −13.55), BCL-XL (−12.28, −17.44, 12.14), MCL-1 (−15.66, 16.19, −1.62) and HK2 (87.0, 15.99, −102.0) were selected. The second stage in the docking experiment corresponded to obtaining the best orientation of a ligand into the active site of a protein, this one treated or selected as a rigid body, through the Lamarckian Genetic Algorithm (LGA) [70]. For this protocol, a population size of 5000 individuals and 50 LGA runs were selected. The best ligand–protein complexes were analyzed according to the potential intermolecular interactions such as hydrogen bonding and the cation–π, π-π stacking. The 2D ligand–protein diagrams were drawn using the software discovery studio visualizer (Dassault Systèmes BIOVIA. Discovery Studio Modeling Environment, Release 2017. Dassault Systèmes; San Diego, CA, USA: 2017). All the alkaloids identified have been submitted to a re-docking process, removing the ligand molecule from the receptor model and then docking it back.

### 4.7. Statistical Analysis

The results of cell viability are shown as the mean ± SD. The statistical significance between the control group and the treatments with the alkaloid fraction were evaluated by analysis of variance (one-way ANOVA) followed by the Newman–Keuls multiple-comparison test, using GraphPad Prism version 5.00 (GraphPad Software, San Diego, CA, USA) data analysis system. In addition, multivariate principal component (PCA) and partial least squares (PLS-DA) statistical analyses were performed using the MUMA package at RStudio (Posit, Boston, MA, USA). The differences were considered significant for *p* ≤ 0.05 and with respect to the control.

## 5. Conclusions

In summary, the present study explored the cytotoxic activity of ten Amaryllidaceae species, characterized by the content of their alkaloids, against six cancer cell lines. The best results (lower cell viability percentages) were obtained for *Crinum jagus* and *Eucharis bonplandii* against the gastric cancer cell line AGS. Our results suggest that the cytotoxic activity might be associated with lycorine and crinine-type alkaloids, which is consistent with previous studies. In addition, this is the first report of phytochemical data for *C. tenera*, *E. formosa*, *P. ventricosa,* and *Z. puertoricensis*. The research focused on evaluating the identified alkaloids in the Bcl-2 protein family (Mcl-1 and Bcl-xL) and HK2, where the in vitro, in silico, and statistical results suggest that powelline and buphanidrine alkaloids could present cytotoxic activity. More tangible evidence is required based on testing the activity of isolated compounds. Finally, combining experimental and theoretical assays allow us to identify and characterize potentially useful alkaloids for cancer treatment.

## Figures and Tables

**Figure 1 molecules-28-02601-f001:**
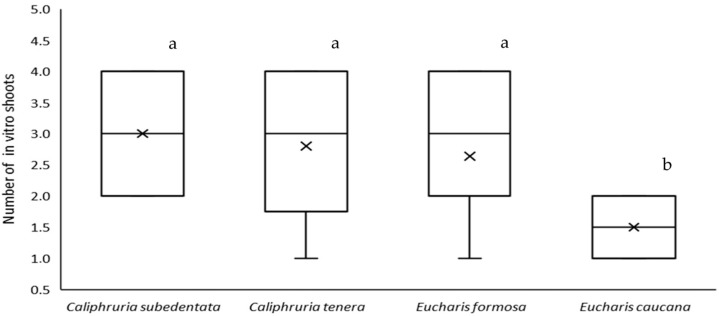
Effect of benzyladenine (BAP, 1.0 mg/L) and naphthaleneacetic acid (NAA, 0.2 mg/L) on the formation of in vitro shoots in *C. subedentata*, *C. tenera*, *E. caucana*, and *E. formosa* explants. According to Tukey’s test, different letters indicate significant differences at the level of (*p* ˂ 0.05).

**Figure 2 molecules-28-02601-f002:**
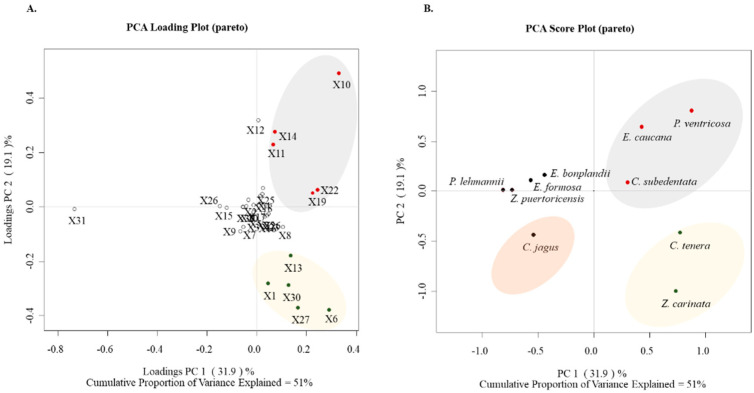
PCA based on the alkaloid profile of alkaloid fractions from Amaryllidaceae species. (**A**): PCA Loading Plot discriminating between alkaloid type and species; (**B**): PCA Score Plot discriminating between total alkaloid profile and species. Table 3 describes the codes assigned to the alkaloids.

**Figure 3 molecules-28-02601-f003:**
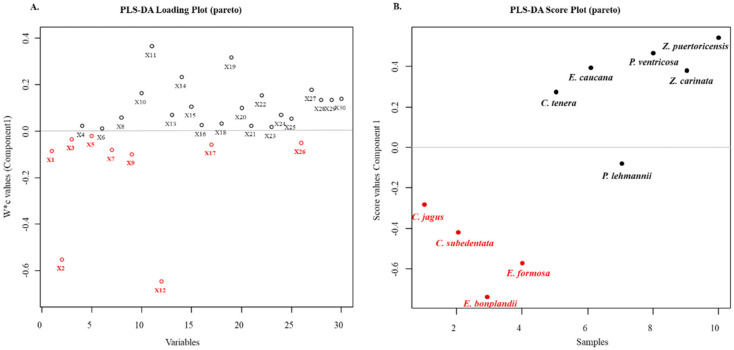
PLS-DA based on alkaloid profile and cell viability in gastric cancer of Amaryllidaceae species. (**A**). PLS-DA loading plot discriminating between alkaloid type and cell viability in gastric cancer cells (AGS). (**B**). PLS-DA score plot discriminating between the alkaloid profile of Amaryllidaceae species and cell viability in gastric cancer cells (AGS). Table 3 describes the codes assigned to the alkaloids.

**Figure 4 molecules-28-02601-f004:**
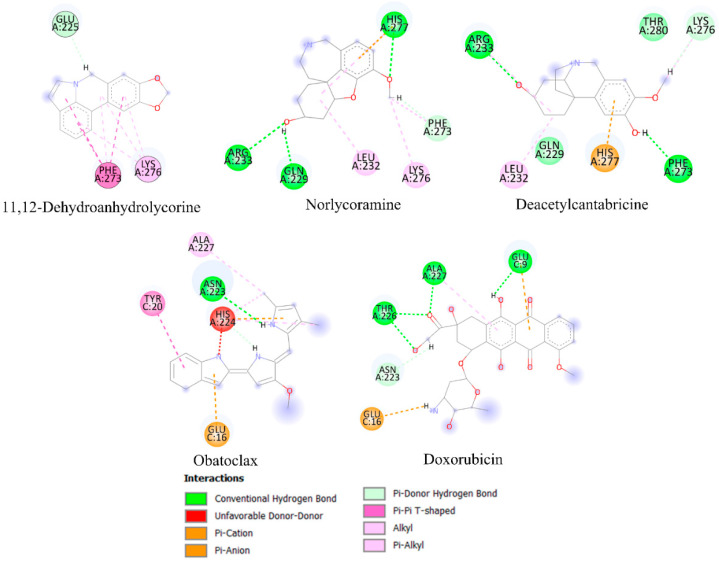
Principal interactions located by the molecular docking experiments between the ligands 11,12-dehydroanhydrolycorine, norlycoramine, and deacetylcantabricine and Mcl-1 protein.

**Figure 5 molecules-28-02601-f005:**
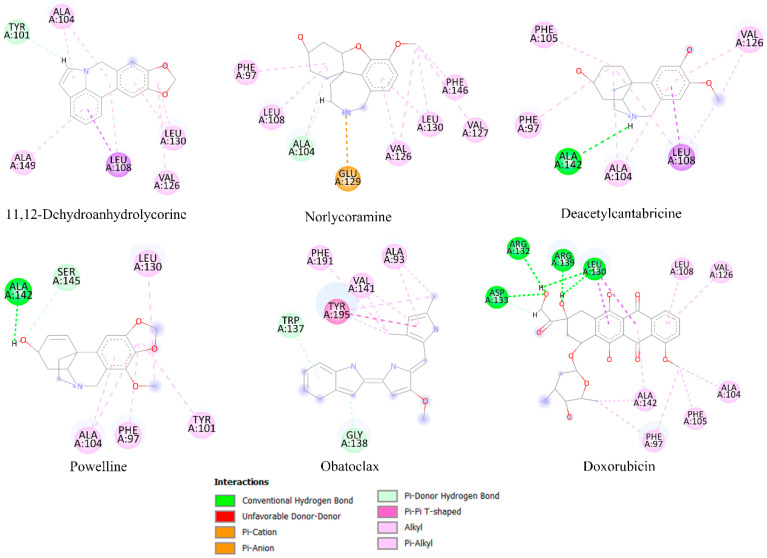
Principal interactions located by the molecular docking experiments between the ligands 11,12-dehydroanhydrolycorine, deacetylcantabricine, norlycoramine, and powelline with Bcl-xL protein.

**Figure 6 molecules-28-02601-f006:**
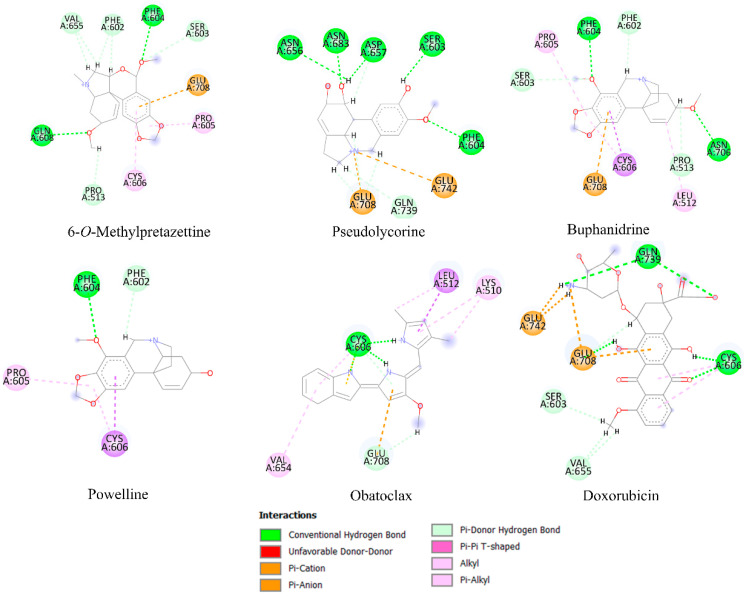
Principal interactions located by the molecular docking experiments between the ligands 6-*O*-methylpretazettine, pseudolycorine, buphanidrine, and powelline with HK-2 protein.

**Table 1 molecules-28-02601-t001:** Geographical origin of the collected Amaryllidaceae species.

Species	Origin Georeference	Voucher
*C. jagus* ^1^	6°9′33.58″ N, 75°21′32.75″ W 2120 m; Marinilla-Antioquia	5270 Alzate
*C. subedentata* ^2^	2°32′27″ N, 76°37′44″ O. 1692 m; Popayan-Cauca	5400 Alzate
*C. tenera* ^2^	2°32′27″ N, 76°37′44″ O. 1692 m; Popayan-Cauca	5400 Alzate
*E. bonplandii* ^1^	4°35′18.47″ N, 75°51′19.78″ W. 1150 m; Armenia-Quindío	5107 Alzate
*E. caucana* ^2^	5°29′54″ N, 76°32′29″ O. 69 m; Lloró, Chocó	5168 Alzate
*E. formosa* ^2^	2°32′27″ N, 76°37′44″ O. 1692 m; Popayan-Cauca	5401 Alzate
*P. lehmannii* ^1^	2°4′27″ N, 76º 54′1′′ O, 1610 m; Rosas-Cauca	5106 Alzate
*P. ventricosa* ^1^	1°12′29″ N,77°27′57″ O, 1663 m; Consaca-Nariño	5402 Alzate
*Z. carinata* ^1^	6°4′30.37″ N, 75°22′46.00″ W, 2150 m; Carmen del Viboral-Antioquia	5307 Alzate
*Z. puertoricensis* ^1^	4°26′54″ N 75°11′56″ O; Ibagué-Tolima	5308 Alzate

^1^ Bulbs from plants collected under field conditions. ^2^ Bulbs from plants collected under field conditions and subsequently micropropagated.

**Table 2 molecules-28-02601-t002:** Cell viability of alkaloid fractions from Amaryllidaceae species.

Species		Cell Viability (%) ^1^					
	AGS	PC3	MCF-7	MDA-MB 231	BT-549	HEC-1B	HaCat
*C. jagus*	48.06 ± 3.35	73.42 ± 3.90	83.91 ± 1.15	58.88 ± 6.15	73.64 ± 3.38	87.65 ± 6.52	91.94 ± 0.83
*C. subedentata*	52.15 ± 2.10	84.29 ± 3.48	98.05 ± 5.53	56.51 ± 0.62	77.67 ± 3.12	88.09 ± 5.90	99.49 ± 0.09
*C. tenera*	73.72 ± 5.90	86.41 ± 2.28	73.94 ± 8.31	58.26 ± 8.12	65.28 ± 2.17	84.54 ± 0.88	99.11 ± 2.76
*E. bonplandii*	45.79 ± 3.05	77.92 ± 3.91	90.61 ± 4.94	57.56 ± 2.99	69.43 ± 5.76	88.25 ± 4.38	94.27 ± 1.28
*E. caucana*	61.46 ± 5.30	80.43 ± 3.79	52.82 ± 2.36	62.35 ± 4.35	65.51 ± 4.24	87.88 ± 4.91	99.57 ± 1.63
*E. formosa*	54.18 ± 2.56	74.02 ± 5.60	94.93 ± 3.41	54.81 ± 4.61	62.36 ± 6.46	80.38 ± 8.46	88.26 ± 0.86
*P. lehmannii*	71.90 ± 11.74	88.54 ± 2.34	73.28 ± 3.75	70.87 ± 4.27	76.40 ± 7.43	87.41 ± 1.49	98.32 ± 2.94
*P. ventricosa*	80.41 ± 6.28	88.91 ± 4.49	98.38 ± 4.28	73.69 ± 4.58	86.11 ± 2.75	83.78 ± 6.63	98.34 ± 2.97
*Z. carinata*	70.33 ± 8.91	88.40 ± 2.10	75.29 ± 7.44	72.46 ± 4.32	62.84 ± 5.37	78.29 ± 1.13	99.55 ± 4.78
*Z. puertoricensis*	67.25 ± 1.90	74.76 ± 5.88	69.86 ± 3.58	56.02 ± 5.63	51.40 ± 1.42	75.95 ± 2.18	97.33 ± 6.69

^1^ Cell viability of the alkaloids fraction against gastric cancer cell line (AGS), prostate cancer cell line (PC3), breast cancer cell lines (MCF-7, MDA-MB 231 and BT-549), uterine cancer cell line (HEC-1B), and human keratinocytes (HaCat) cells at 30 µg/mL. The experiments were performed in triplicate. Data are presented as the mean ± SD from three separate experiments.

**Table 3 molecules-28-02601-t003:** Alkaloid profile of alkaloid fractions from Amaryllidaceae species.

Species ^1^		Alkaloids Type (mg/gDW) ^2^						
	Crinane		Galanthamine		Lycorine		Miscellaneous	
*C. jagus*	Crinine (X1) Crinane acetate (X3) Buphanidrine (X5) Crinamine (X7) Powelline (X9)	0.073 0.006 0.002 0.033 0.050			5,6-Dihydrobicolorine (X13) Anhydrolycorine (X15) 11,12-Dehydroanhydrolycorine (X19) Lycorine (X31)	0.009 0.090 0.022 0.295	Trisphaeridine (X22)	0.026
*C. subedentata*	Crinine (X1) 8-*O*-Demethylmaritidine (X2) 6-*O*-Methylpretazettine (X4) Haemantamine (X6) Tazettine (X8) Hamayne (X10)	0.216 1.700 0.022 1.005 0.211 0.771	Galanthamine (X12)	1.824	5,6-Dihydrobicolorine (X13) Anhydrolycorine (X15) Assoanine (X20) 11,12-Dehydroanhydrolycorine (X19) Lycorine (X31)	0.069 0.279 0.038 0.210 0.629	Ismine (X21) Trisphaeridine (X22)	0.032 0.140
*C. tenera*	Crinine (X1) 8-*O*-Demethylmaritidine (X2) Haemantamine (X6) Tazettine (X8) Hamayne (X10)	0.133 0.282 1.114 0.010 0.377	Galanthamine (X12)	0.055	5,6-Dihydrobicolorine (X13) 11,12-Dehydrolycorene (X16) 11,12-Dehydroanhydrolycorine (X19) Lycorine (X31) Pseudolycorine (X26)	0.372 0.020 0.312 0.167 0.009	Trisphaeridine (X22) Demethylismine (X23)	0.371 0.018
*E. bonplandii*	Crinine (X1) 8-*O*-Demethylmaritidine (X2) Hamayne (X10)	0.018 0.339 0.066	Galanthamine (X12) Sanguinine (X14) Epinorgalanthamine (X17)	0.280 0.004 0.006	5,6-Dihydrobicolorine (X13) Anhydrolycorine (X15) 11,12-Dehydroanhydrolycorine (X19) Lycorine (X31) Pseudolycorine (X26)	0.003 0.056 0.008 0.581 0.004	Trisphaeridine (X22)	0.008
*E. caucana*	Crinine (X1) 8-*O*-Demethylmaritidine (X2) 6-*O*-Methylpretazettine (X4) Tazettine (X8) Hamayne (X10)	0.013 0.015 0.012 0.045 0.324	Galanthamine (X12) Sanguinine (X14) Narwedine (X18) *N*-formylnorgalanthamine (X25)	0.181 0.434 0.010 0.027	Anhydrolycorine (X15) 11,12-Dehydroanhydrolycorine (X19) Lycorine (X31)	0.025 0.065 0.135	Ismine (X21) Galanthindole (X24)	0.008 0.016
*E. formosa*	8-*O*-Demethylmaritidine (X2) Haemantamine (X6) Tazettine (X8) Hamayne (X10)	0.068 0.315 0.019 0.247	Galanthamine (X12) Sanguinine (X14)	0.921 0.012	5,6-Dihydrobicolorine (X13) Lycorine (X31)	0.010 2.390	Ismine (X21) Trisphaeridine (X22)	0.004 0.032
*P. lehmannii*	8-*O*-Demethylmaritidine (X2)	0.015	Galanthamine (X12) Sanguinine (X14)	0.015 0.004	Lycorine (X31) Pseudolycorine (X26)	0.090 0.029		
*P. ventricosa*	Hamayne (X10) Deacetylcantabricine (X11)	0.036 0.018	Galanthamine (X12)	0.015	11,12-Dehydroanhydrolycorine (X19)	0.013	Trisphaeridine (X22)	0.017
*Z. carinata*	Crinine (X1) 8-*O*-Demethylmaritidine (X2) Haemantamine (X6) Tazettine (X8)	0.718 0.138 1.707 0.417	Galanthamine (X12) Lycoramine (X27)	0.354 2.274	5,6-Dihydrobicolorine (X13) 11,12-Dehydrolycorene (X16) Anhydrolycorine (X15) 11,12-Dehydroanhydrolycorine (X19) Galanthine (X30)	0.269 0.011 0.197 0.450 1.380	Ismine (X21) Galanthindole (X24) Trisphaeridine (X22)	0.075 0.105 0302
*Z. puertoricensis*	Deacetylcantabricine (X11) Aulicine (X28)	0.021 0.006	Sanguinine (X14) Norlycoramine (X29)	0.003 0.006	Anhydrolycorine (X15) Assoanine (X20) 11,12-Dehydroanhydrolycorine (X19) Lycorine (X31)	0.030 0.004 0.006 0.099		

^1^ Alkaloidal fraction from the bulbs of each species. ^2^ Quantitative values obtained by response factor using codeine as internal standard (mg of alkaloid per g of dry weight).

**Table 4 molecules-28-02601-t004:** Molecular docking analysis.

		Estimated Free Energy of Binding ^1^			
Alkaloid Type	Compound	Bcl-2	Mcl-1	Bcl-xL	HK2
Lycorine	5,6-dihydrobicolorine	−5.65	−4.18	−6.46	−6.69
	Anhydrolycorine	−5.79	−5.01	−6.23	−6.83
	Assoanine	−5.63	−5.02	−5.66	−6.62
	Lycorine	−5.75	−3.85	−4.77	−8.25
	11,12-Dehydroanhydrolycorine	−5.57	−5.58	−6.79	−6.69
	Pseudolycorine	−5.79	−3.51	−4.24	−8.48
Crinane	8-*O*-demethylmaritidine	−6.40	−3.62	−4.71	−7.61
	Buphanidrine	−6.35	−4.05	−5.04	−8.59
	Crinamin	−5.80	−4.32	−5.95	−7.64
	Crinane acetate	−6.55	−4.34	−5.15	−8.01
	Crinine	−6.00	−4.78	−5.33	−8.26
	Haemanthamine	−5.70	−4.71	−5.50	−7.75
	Hamayne	−6.02	−3.57	−4.83	−7.66
	Powelline	−6.10	−3.98	−6.93	−8.80
	Tazettine	−6.53	−4.74	−4.74	−7.76
	6-*O*-Methylpretazettine	−6.53	−3.31	−4.64	−8.38
	Deacetylcantabricine	−5.88	−5.82	−6.76	−6.90
	Aulicine	−5.46	−4.95	−6.20	−7.72
Galanthamine	Galanthamine	−5.78	−3.67	−4.91	−7.23
	Sanguinine	−5.87	−3.84	−5.16	−7.54
	Epinorgalanthamine	−6.56	−4.63	−6.64	−7.87
	Narwedine	−6.34	−4.16	−5.46	−7.65
	*N*-formylnorgalanthamine	−5.64	−4.83	−6.23	−6.01
	Norlycoramine	−6.27	−5.39	−6.99	−7.63
Miscellaneous	Ismine	−5.48	−3.67	−4.01	−6.43
	Trisphaeridine	−5.69	−4.57	−5.55	−6.41
	Galanthindole	−5.57	−4.54	−4.89	−7.03
Control	Obatoclax ^2^	−6.76	−4.62	−5.77	−7.64
	Doxorubicin ^2^	−5.66	−3.48	−6.74	−8.35

^1^ The energy values are expressed in kcal/mol. ^2^ Obatoclax and Doxorubicin compounds have been tested as positive controls.

## Data Availability

Not applicable.

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
