# Peer review of "Cytotoxic Activity of Amaryllidaceae Plants against Cancer Cells: Biotechnological, In Vitro, and In Silico Approaches"

_molecules, 2023, doi:10.3390/molecules28062601_

Round 1

Reviewer 1 Report

Title, replace in cancer cells with against cancer cells

Clear the aim of study in the abstract

Add some data

Brief the first paragraph of introduction and clarify the aim at the end of introduction

Pay attention for linguistically and structural errors

Line 109, use the full scientific name for the frist time

Clear the sample size

Provide the origin and model of all devices

Enhance the statical analysis part and clear the pre-tests (homogeneity,) used to analyze your data with ANOVA

indicate the lowercase letters in the table footnotes and do that in all tables and figures

provide images for cancer cell lines and the effects of alkaloids on them

the data is poorly presented and clear the part of in silico

some tables not statistically analysed

check the outputs and scientific names in references

Author Response

The answers to the comments and questions are in the attached document

Reviewer 2 Report

Dear Editors and Authors:

The authors characterized the cytotoxic activities of alkaloids from Amaryllidaceae species in six cell lines. However, the experiment design and writing were poor(no positive control), and the results were hard to understand. Based on the criteria of the Journal of Molecules, I suggested reconsidering after major revision.

Major concern /comments:

.

1.The study design was poor and no positive control. The foundational data of the manuscript is Table 2—cell viability of alkaloid fractions from Amaryllidaceae species. But the study design was poor in this experiment. E.g. where the authors got the six cell lines E.g. ATCC? Why the authors chose the 30 µg/mL concentration in the cell viability studied? Where is the positive control drug?

2. Please clarify why the authors chose the Bcl-2 protein family (Mcl-1 and Bcl-xL) and HK2 in the silico analysis?

3. “Our results suggest that the cytotoxic activity might be associated with lycorine and crinine-type alkaloids, which is consistent with previous studies.” Please clarify the innovation of the study.

Author Response

(The authors gave the same response as above.)

Reviewer 3 Report

Dear Editor

Thank you for allowing me to review this manuscript. The manuscript by Trujillo et al. studied the Cytotoxic activity of Amaryllidaceae plants against six cancer cell lines AGS, BT- 18 549, HEC-1B, MCF-7, MDA-MB 231, and PC3, with HaCat as a control cell line.

The search is generally good, but it needs some modifications. I have listed them below:

Abstract:

Authors should write the meaning of each abbreviation in the first mention as cell lines.

Introduction:

The introduction is long, and the information about the plants under study is very scarce. Please provide the introduction section with more information about tested materials.

Results:

Part of plant materials is good, but the abbreviation of naphthaleneacetic acid in lines 130 and 491 is NAA not ANA.

Part of Cytotoxic activities of the different tested alkaloid fractions: The authors expressed the anti-cancer effect of the tested alkaloids by the inhibition percent at 30 μg/ml, which is good, but the perfect expression by IC50.

Part of Alkaloid profile of the different alkaloid fractions is good.

Part of Molecular docking analysis:

The alkaloidal fraction of E. bonplandii showed the best activity, however, the identified compounds in this fraction did not show good binding activity with the tested proteins; Why did not the authors choose other proteins for docking studies that might theoretically rationalize this activity? 

What about validation of the docking studies? 

3D representation of the results may increase the usefulness of the research. 

Discussion: 

In the discussion, the author must mention the comparison with other relevant studies.  Discussion is long and did not discus the mode of action of these alkaloids.

Materials and methods:

The author did not refer to the method used to alkaloids extraction. Please mention the reference.

The author did not refer to the method used to identify the alkaloid compounds responsible for the biological activity mentioned in the abstract. The technique used to determine them should be mentioned. 

The author did not refer to the method used to anticancer.  Please mention the reference. 

Author Response

(The authors gave the same response as above.)

Round 2

Reviewer 1 Report

Now can be accepted

Author Response

The authors thank the reviewer

Reviewer 2 Report

Major concern:

1.The study design was poor. Why the authors chose the 30 µg/mL concentration in the cell viability studied? The authors explained that :The cells were treated with the alkaloid fraction at a concentration of 30 µg/mL, according to the cytotoxic activity criteria for a crude extract as established by the American National Cancer Institute (NCI) in IC50 < 30 µg/mL. . But I searched on the NCI website , no related cytotoxic activity criteria for a crude extract , most published paper cited the “Suffness, M.; Pezzuto, J.M. Assays related to cancer drug discovery. In Methods in Plant Biochemistry: Assays for Bioactivity; Hostettmann, K., Ed.; Academic Press: London, UK, 1990; pp. 71–133.” I think some people make up the cytotoxic activity criteria. Please clarify of  experiment design again.

2.The results were not reliable. The authors added the IC50 data of positive control and Amaryllidaceae alkaloids. But in the published papers, the  IC50 of lycorine and haemanthamine were(IC50 < 0.5 µM) and(IC50 = 7.5 µM at 24 and 48 h)  in AGS cells(Cytotoxicity and Antiviral Properties of Alkaloids Isolated from Pancratium maritimum), and the authors IC50 value were 1000-folg higher than the published (haemanthamine (1.12 ± 0.35 µg/mL) and lycorine (0.62 ± 0.13 µg/mL)).,  and the IC50 of positive control doxorubicin  (5.73 ± 0.80 µg/mL) was comparable to the crude extract 30ug/ml. These results were abnormal. Please provide the picture of your cell line and the MTT results of positive control and Amaryllidaceae alkaloids, especially check the results.

Author Response

Cytotoxic activity of Amaryllidaceae plants against cancer cells: biotechnological, in vitro, and in silico approaches

List of responses to reviewer comments:

Comment: The study design was poor. Why the authors chose the 30 µg/mL concentration in the cell viability studied? The authors explained that: “The cells were treated with the alkaloid fraction at a concentration of 30 µg/mL, according to the cytotoxic activity criteria for a crude extract as established by the American National Cancer Institute (NCI) in IC50 < 30 µg/mL. But I searched on the NCI website, no related cytotoxic activity criteria for a crude extract, most published paper cited the “Suffness, M.; Pezzuto, J.M. Assays related to cancer drug discovery. In Methods in Plant Biochemistry: Assays for Bioactivity; Hostettmann, K., Ed.; Academic Press: London, UK, 1990; pp. 71–133.” I think some people make up the cytotoxic activity criteria. Please clarify of experiment design again.

Response: In this new version, we tried to clarify the choice of this concentration in the study. The cytotoxic activity criterion for a crude extract established by the American National Cancer Institute (NCI) at IC50 < 30 µg/mL has recently been used in research as a screening parameter (1)(2). This criterion makes it possible to reduce the number of experiments in preliminary tests to determine IC50 values. In addition, two references are added where the use of such criteria is used. The following sentence has been included in the text:

Results were expressed as the percentage of cell viability at 30 µg/mL, considering the American National Cancer Institute (NCI) guidelines for a promising crude extract, which states that a crude extract with IC50 less than 30 μg/mL is a candidate for chemical characterization [29,30].

Comment: The results were not reliable. The authors added the IC50 data of positive control and Amaryllidaceae alkaloids. But in the published papers, the IC50 of lycorine and haemanthamine were (IC50 < 0.5 µM) and (IC50 = 7.5 µM at 24 and 48 h) in AGS cells (Cytotoxicity and Antiviral Properties of Alkaloids Isolated from Pancratium maritimum), and the authors IC50 value were 1000-folg higher than the published (haemanthamine (1.12 ± 0.35 µg/mL) and lycorine (0.62 ± 0.13 µg/mL))., and the IC50 of positive control doxorubicin (5.73 ± 0.80 µg/mL) was comparable to the crude extract 30 ug/ml. These results were abnormal. Please provide the picture of your cell line and the MTT results of positive control and Amaryllidaceae alkaloids, especially check the results.

Response: The results of cell viability in AGS gastric cancer cells for lycorine and haemanthamine (Table 1) show large differences in the IC50 values reported for lycorine (< 0.5 µM) and haemanthamine (7.5 µM at 24 and 48 h) by Masi et al (4). These variations are directly associated with determining parameters in cell culture as number of cell pass, culture medium, number of cells in the experiment and incubation time. In addition, modifications in the method for determining cell viability (MTT) significantly affect the results. In this case, one of the changes observed in the report by Masi et al. is the solubility of the formazan crystals with acidic isopropanol. Where it has been reported that this solvent generates a low absorbance at 570 nm and therefore an instability. For this reason, the method suggests the use of DMSO to achieve greater reproducibility and reliability of the results (5).

Although IC50 values for lycorine (4.17 µg/mL or 14.51 µM) and haemanthamine (13.18 µg/mL or 43.74 ± 1.56 μM) are higher than those reported by Masi et al., a broader comparison with other authors is needed. However, reports of cellular viability of Amaryllidaceae alkaloids in gastric cancer cells AGS is scarce. Therefore, it is not possible at this time to extend the range of comparison of the results obtained.

In this study, cell viability results for lycorine and haemanthamine were obtained at concentrations of 0.1, 1.0, 10, 20, 30, 50 and 100 μg/mL (Figure 1) with incubation for 48 h. However, the existing report for lycorine shows that the IC50 value was determined at a lower concentration range (0.5, 1.0, 5.0, 10 and 50 μM) with incubation for 48 h. These differences in the concentration range do not allow an accurate comparison of IC50 values. Regarding haemanthamine, the reported data are unclear because they do not show 48 h incubation results for this molecule.

Doxorubicin was selected as a positive control due to existing reports as a chemotherapeutic agent in gastric cancer. Results with doxorubicin showed that the concentrations evaluated (0.31, 0.62, 1.25, 2.5, 5 and 10 μg/mL) allowed an IC50 = 5.73 μg/mL with 48 h incubation. This result shows that the determined concentration depends on the incubation time because an IC50 = 15 μg/mL with 24 h incubation has been reported in AGS cells (3).

The IC50 concentrations were corrected in the document for lycorine and haemanthamine, because the previously reported value corresponds to the log IC50 and not to the IC50.

Table 1. IC50 for Amaryllidaceae alkaloids and positive control.

Log IC50 (µg/mL)

IC50 (µg/mL)

Lycorine

0.62

4.17

Haemanthamine

1.12

13.18

Doxorubicin

0.76

5.73

Figure 1. Cell viability for lycorine, haemanthamine and doxorubicin in AGS cells. Cell viability was determined by MTT assay with 48 hours incubation.

Reference

  1. Botteon, C.E.A., Silva, L.B., Ccana-Ccapatinta, G.V., Silva, T.S, Ambrosio, S.R., Veneziani, R.C.S., et al. Biosynthesis and characterization of gold nanoparticles using Brazilian red propolis and evaluation of its antimicrobial and anticancer activities. Sci Rep. 2021, 11, 1–16.
  2. Omoruyi, S.I., Kangwa, T.S., Ibrakaw, A.S., Cupido, C.N., Marnewick, J.L., Ekpo, O.E., et al. Cytotoxic activities of selected plants of the family Amaryllidaceae on brain tumour cell lines. South African J Bot. 2021, 136, 118-125.
  3. Firouzi Amoodizaj, F., Baghaeifar, S., Taheri, E., Sefidan Jadid M.F., Safi, M., Seyyed Sani N., et al. Enhanced anticancer potency of doxorubicin in combination with curcumin in gastric adenocarcinoma. J Biochem Mol Toxicol. 2020, 34, 22486.
  4. Masi, M., Di Lecce, R., Mérindol, N., Girard, M.P., Berthoux, L., Desgagné-Penix, I., et al. Cytotoxicity and Antiviral Properties of Alkaloids Isolated from Pancratium maritimum. Toxins (Basel). 2022,14, 262.
  5. Babacan, Ü., Kaba, A., Akçakale, F., Cengiz, M.f., Akinci, E. Optimization of some parametric values of MTT for the determination of human melanoma (SK-Mel-30) cell viability. Int J Life Sci Biotechnol. 2021, 5, 9–20.
